# Vaccine-Induced Immune Thrombocytopenia and Thrombosis (VITT)—Insights from Clinical Cases, In Vitro Studies and Murine Models

**DOI:** 10.3390/jcm12196126

**Published:** 2023-09-22

**Authors:** Venkata A. S. Dabbiru, Luisa Müller, Linda Schönborn, Andreas Greinacher

**Affiliations:** Institut für Transfusionsmedizin, Universitätsmedizin Greifswald, 17489 Greifswald, Germany; venkata.dabbiru@med.uni-greifswald.de (V.A.S.D.); luisa.mueller@med.uni-greifswald.de (L.M.); linda.schoenborn@med.uni-greifswald.de (L.S.)

**Keywords:** VITT, HIT, platelet factor 4, anti-PF4 antibodies, ChAdOx1 nCoV-19, Ad26-COV-2S

## Abstract

An effective worldwide vaccination campaign started and is still being carried out in the face of the coronavirus disease 2019 (COVID-19) pandemic. While vaccines are great tools to confront the pandemic, predominantly adenoviral vector-based vaccines can cause a rare severe adverse effect, termed vaccine-induced immune thrombocytopenia and thrombosis (VITT), in about 1 in 100,000 vaccinated individuals. VITT is diagnosed 5–30 days post-vaccination and clinically characterized by thrombocytopenia, strongly elevated D-dimer levels, platelet-activating anti-platelet factor 4 (PF4) antibodies and thrombosis, especially at atypical sites such as the cerebral venous sinus and/or splanchnic veins. There are striking similarities between heparin-induced thrombocytopenia (HIT) and VITT. Both are caused by anti-PF4 antibodies, causing platelet and leukocyte activation which results in massive thrombo-inflammation. However, it is still to be determined why PF4 becomes immunogenic in VITT and which constituent of the vaccine triggers the immune response. As VITT-like syndromes are increasingly reported in patients shortly after viral infections, direct virus-PF4 interactions might be most relevant. Here we summarize the current information and hypotheses on the pathogenesis of VITT and address in vivo models, especially murine models for further studies on VITT.

## 1. Introduction

The coronavirus disease 2019 (COVID-19) pandemic caused by severe acute respiratory syndrome coronavirus 2 (SARS-CoV-2) had a major impact on all parts of the world. Concomitant with efforts to control virus transmission, vaccination is an indispensable tool to contain the pandemic. Different types of vaccines were developed rapidly, including adenoviral vector-based vaccines, namely ChAdOx1 nCoV-19 (Vaxzevira/AstraZeneca), Ad26-COV-2S (Janssen/Johnson and Johnson), Gam-COVID-Vac (Sputnik V), along with mRNA vaccines such as BNT162b1 (Pfizer/BioNTech), mRNA-1273 (Moderna) and whole inactivated virus vaccines BBV152 COVAXIN (Bharat Biotech) and PiCoVacc (Sinovac) [1]. Predominantly adenoviral vector-based vaccines rendered unexpected, rare [2] adverse effects in a small fraction of recipients, who developed thrombocytopenia and thrombosis at unusual sites [3,4,5]. Several hundred cases have been reported during mass vaccinations worldwide. This condition is clinically termed “thrombosis and thrombocytopenia syndrome (TTS)”, while “vaccine-induced immune thrombocytopenia and thrombosis (VITT)” is reserved for the subset of patients who develop TTS in the presence of high titer, platelet-activating anti-platelet factor 4 (PF4) antibodies [6]. Following the UK Haematology Expert Group consensus [7] and World Health Organization (WHO) recommendations [6], VITT typically occurs five to thirty days post-vaccination and is clinically characterized by thrombocytopenia, strongly elevated D-dimer levels, the ability of anti-PF4 antibodies to activate platelets and the occurrence of thrombosis in atypical sites such as cerebral venous sinus and/or splanchnic veins [8].

VITT remains a risk for patients in low- and middle-income countries with an ongoing COVID-19 vaccination campaign that can only afford adenoviral vector-based vaccines [9]. Furthermore, adenoviral vectors are providing a promising basic tool for the development of new vaccines for neglected diseases, which are affordable for low- and middle-income countries. In addition to COVID-19 vaccination, the first clues point to the possibility that not only adenoviral vector-based vaccines can cause VITT-like symptoms [10,11,12,13]. For this reason, a thorough understanding of underlying pathological mechanisms is important to make vaccination safer and to identify and treat patients affected by VITT-like syndromes.

A related prothrombotic disorder that involves anti-PF4 antibodies is heparin-induced thrombocytopenia (HIT). HIT has been known since the 1960s [14,15]. HIT and VITT show similar pathophysiology and clinical manifestations. While in HIT the pathologic trigger has clearly been defined (various forms of polyanions, mostly heparin), the exact pathological trigger of VITT still remains elusive. A systematic understanding of HIT and its variants enabled translating knowledge obtained in HIT to VITT. After a brief comparison between VITT and HIT, we summarize the current information on VITT based on clinical cases, in vitro studies and murine models and provide some hypotheses on their pathogenesis.

## 2. Similarities and Differences between VITT, HIT and Its Variants

With the onset of VITT and recognition of the causative role of pathogenic anti-PF4 immunoglobulin G (IgG) antibodies [16], PF4-dependent disorders gained major attention. Anti-PF4 antibodies develop within a short time range of five to ten days after exposure to heparin or vaccine, in HIT [17,18] and VITT [3,8], respectively. This strongly indicates that both conditions arise as a secondary immune response. Also, in both HIT [19,20] and VITT [16], anti-PF4 antibodies cause FcγIIa (CD32a) receptor-mediated activation of platelets and other immune cells.

While similarities between HIT and VITT are confined to their PF4-mediated immunological responses, differences between them stand out and are relevant for diagnosis and treatment [21]. Classic HIT is dependent on heparin, where PF4-heparin complexes induce an immune response resulting in high titer anti-PF4-heparin antibodies. The formation of these PF4-heparin complexes is of utmost importance in HIT; therefore, the text always refers to these complexes in HIT, whilst for VITT, if not stated otherwise the reactivity to PF4 alone is described.

Central to the formation of PF4-heparin complexes is the positively charged ring of lysine and arginine amino acids at the equatorial plane of the PF4 tetramer, facilitating electrostatic binding of the negatively charged heparin molecule [22,23] (Figure 1a). This causes a conformational change in PF4 [24], resulting in a new epitope at the polar regions of the tetrameric PF4 molecule. Anti-PF4-heparin antibodies bind to this neoepitope at the polar regions (Figure 1b, upper panel). On the contrary, VITT is heparin-independent. Based on alanine scanning mutagenesis, Huynh et al. identified the epitopes for anti-PF4 antibody binding in the equatorial plane of PF4, overlapping the heparin binding sites [25]. Heparin, as well as heparin-independent antibodies to PF4, bind at the same region on PF4. Whether PF4 is changed in its structure in the region of the heparin-binding site after complex formation of PF4 with currently still unknown constituents of the vaccine in VITT is currently under investigation. In line with the strong but transient immune response to PF4 observed in VITT, such conformational changes might be likely, as outlined below in the chapter on post-transfusion purpura, which is another example in medicine in which an allo-immune response can become an autoimmune response by epitope spreading.

Furthermore, recent studies suggested the subsequent competitive nature of heparin and VITT antibodies in binding to PF4 [25,26] (Figure 1b, lower panel). Thereby heparin inhibits platelet activation by VITT antibodies by competing with the VITT antibody binding site [26]. Thus, heparin might even ameliorate VITT [27,28], unlike in HIT where it is the driving force of the disease. Consistently, a meta-analysis of the WHO of published case series found no increase in adverse outcomes in VITT patients treated with heparin in comparison to patients treated with non-heparin anticoagulants, intravenous immunoglobulin (IVIG) and steroids [6]. Nevertheless, caution might still be required as some VITT patients possess anti-PF4 antibodies that cross-react with PF4-heparin complexes. This was especially observed in the sera of VITT patients that also showed reactivity in chemiluminescent HIT assays [29]. Therefore, it was advised to administer non-heparin anticoagulants in initial stages of the disease [30] until the induction of platelet activation by patient sera in the presence of heparin has been excluded. If non-heparin anticoagulants are not available or affordable, especially in low- and middle-income countries, heparin might still pose a valid option to treat VITT patients, as a delay in anticoagulation therapy would be more dangerous than the low risk of the presence of rare antibodies in VITT which are enhanced by heparin [6].

Interestingly, earlier reports [31,32] indicate a certain heterogeneity in the antibodies of HIT patients. The authors purified anti-PF4 and anti-PF4-heparin antibodies from the sera of HIT patients and found that some HIT patients also had anti-PF4 antibodies besides anti-PF4-heparin antibodies. As demonstrated by serotonin release assays, the strongest platelet activation was seen when purified anti-PF4-heparin antibodies were incubated with platelets in the presence of heparin. However, at the time of these studies, it was not known that the addition of PF4 can enhance the reactivity of anti-PF4 antibodies. Recently we found that 30% of HIT patients also have anti-PF4 antibodies [29]. Some of these antibodies strongly activate platelets in the presence of PF4. An important future research question is whether the presence of anti-PF4 antibodies besides anti-PF4-heparin antibodies impacts outcomes of patients with HIT.

To better understand the prevalence of HIT-like and VITT-like antibodies, the emphasis lies now on developing laboratory tools to distinguish between the two diseases, ideally based on differences in binding to different epitopes on PF4. A first approach (fluid-phase enzyme immunoassay) has been introduced by Warkentin et al., showing the relevance of PF4 conformation in diagnostic differentiation between VITT and HIT [33].

In autoimmune HIT (aHIT), first described in 2001 [34], patients develop anti-PF4 antibodies in response to heparin in addition to heparin-independent antibodies. Once these antibodies are formed, they can bind to PF4 also in the absence of heparin. By binding to PF4, these antibodies induce a conformational change in PF4 similar to the conformational change induced by heparin, which then allows binding of the heparin-dependent antibodies [23,35]. In 2008 [36], some patients were described who developed heparin-independent pathogenic anti-PF4 antibodies [37]. The stimulus for antibody generation could be complex formation of PF4 with polyanions other than heparin, such as highly negatively charged molecules on bacterial surfaces, platelet-derived chondroitin sulfate [38], polyphosphates [39] or nucleic acids released from endothelial cells during knee replacement surgery [40]. A subset of these antibodies may activate platelets after binding to PF4 alone, possibly due to epitope spreading, i.e., in aHIT [35].

Antibodies in aHIT have been extensively investigated by biophysical techniques such as single molecule-force spectroscopy and isothermal titration calorimetry. They show that the binding forces of anti-PF4 antibodies differ between the different types of antibodies found in classical HIT and aHIT [35]. Those antibodies, which are only positive by ELISA but do not activate platelets, have the lowest binding force. Heparin-dependent anti-PF4 antibodies have an intermediate binding force, and anti-PF4 antibodies which activate platelets independently of heparin have the highest binding force [35]. PF4 molecules repel each other due to their strong positive charge, measured as Zeta potential (Figure 2a,b). When negatively charged heparin binds to PF4 it neutralizes the repellent forces, allowing a close approximation of two PF4 molecules. This results in the fusion of charge clouds of two PF4 molecules and energy release. This energy induces conformational changes in PF4 (Figure 2a, middle panel). This conformational change then expresses the neoantigen to which heparin-dependent antibodies bind (Figure 2a, right panel). The anti-PF4 antibodies in aHIT have such a high binding affinity to PF4 that they can also force PF4 molecules into close proximity with consecutive fusion of the charge clouds and conformational change (Figure 2b, middle panel). This facilitates the formation of multimolecular complexes consisting of PF4 and these antibodies (Figure 2b, right panel). The heparin-independent anti-PF4 antibodies in VITT also seem to have a very high binding affinity to PF4. They also cluster PF4, but their biophysical characteristics remain to be investigated.

## 3. What Explains the Secondary Immunological Response in VITT?

A crucial characteristic of VITT is a very high titer of IgG anti-PF4 antibodies in patients as early as five days after vaccination [4,5,41] that can only be explained by a secondary immune response. This poses the question: what is the primary immune trigger? A recent hypothesis [42] for the trigger of the primary immune response in HIT emphasizes that potentially an evolutionarily conserved, charge-based feature of innate immunity is the underlying cause. The surface negative charge of bacteria and other pathogens had evolutionarily sensitized the innate immune system to respond through mechanisms such as the activation of the complement system [42] and production of cationic antimicrobial peptides [43]. Accordingly, the positively charged chemokine PF4 is yet another component in the arsenal of immunity, bridging between the innate and the adaptive immune system. PF4 binds to most bacterial species by charge-based electrostatic interactions and subsequently induces production of anti-PF4 antibodies, which in turn opsonize PF4–bacterial complexes followed by phagocytosis [44]. A preceding exposure to one infectious agent (Figure 3a) could induce the rapid generation of anti-PF4 IgG antibodies from preformed B-cells during infection with another bacterial species, as a result of an interaction between PF4 with the second infectious agent (Figure 3b, upper and middle panels). This secondary immune response allows the immune system to efficiently tackle pathogens with IgG antibodies without any previous encounter within a short period of time [44]. Analogous to HIT, a theory to explain the secondary immune response in VITT is an induction of conformational changes in PF4 through electrostatic interactions with so far not clearly defined vaccine components [16,45] and neoepitope formation (Figure 3b, lower panel). This is followed by the activation of preformed B-cells to produce high titer antibodies (Figure 3b, middle panel). However, this interaction must be different from the interaction between bacteria and PF4 leading to HIT antibodies, as HIT and VITT anti-PF4 antibodies bind to different epitopes on PF4.

The easiest approach to identify the binding partner of PF4 in the vaccine that induces neoepitope formation is to allow the potential binding partners to interact with PF4 and evaluate the binding capacity of anti-PF4 antibodies with such complexes. However, this strategy cannot be carried out in VITT, because the boosted immune response in VITT triggers anti-PF4 antibodies with such high binding avidity that they bind to PF4 even without the necessity of a cofactor [46] (Figure 4a). In fact, in the PF4-induced platelet activation assay, anti-PF4 antibodies of many VITT patients activate platelets without any cofactors [47]. By dynamic light scattering (DLS) it was shown that these antibodies even form large complexes with PF4 in the absence of any cofactor. This phenomenon of a broader activity when antibodies are strongly boosted is also known as epitope spreading [48]. In hematology this is well described for the rare adverse transfusion reaction posttransfusion purpura (PTP). Alloantibodies are causative against a specific blood group antigen on platelets, human platelet antigen 1a (HPA-1a). One amino acid exchange, i.e., the presence of either Leucine or Proline at amino acid residue 33 on the integrin chain ß3 (GPIIIa), causes either the blood group HPA-1a or HPA-1b [49]. Women with the blood group HPA-1b/b, who have been immunized against the platelet alloantigen HPA-1a during pregnancy, can develop high titer anti-HPA-1a alloantibodies when they receive a blood transfusion from an HPA-1a donor many years later, e.g., during elective surgery [50] (Figure 4b, upper panel). About five to fourteen days after transfusion of the HPA-1a incompatible blood product, they develop severe thrombocytopenia and bleeding [51,52]. This is caused by depletion of their own HPA-1a negative platelets by the boosted anti-HPA-1a alloantibodies. These anti-HPA-1a antibodies lost their specificity by epitope spreading and bind also to the patient’s autologous platelets, although these platelets do not even express HPA-1a [50] (Figure 4b, lower panel).

In HIT the causative role of heparin had been first inferred by the clinical observation that HIT always occurs after heparin application, but only experimentally demonstrated by functional assays and ELISA, showing that the antibodies bind primarily to PF4-heparin complexes [32]

However, if only patients with heparin-independent aHIT antibodies would have been available, the role of heparin could not have been proven experimentally, as aHIT antibodies bind to PF4 independent of its cofactor heparin. The current dilemma in characterizing the trigger for the VITT anti-PF4 antibodies is that the antibodies bind to PF4 independent of any cofactor, most likely by the above described phenomenon of epitope spreading.

## 4. Insights into Vaccine Components and Their Mechanistic Role in Thrombo-Inflammation of VITT

Hemostatic and inflammatory processes are evolutionarily conserved mechanisms known to act synergistically after an insult such as infection or tissue injury [53]. Thrombo-inflammation is a detrimental phenomenon where thrombosis and inflammation occur in the microvasculature as a response to various stimuli, such as pathogens, tissue injury and inflammatory danger signals [53,54]. Thrombo-inflammation has been reported in a wide variety of diseases including, but not limited to, sepsis and ischemia-reperfusion injury [53], dengue infection [55], COVID-19 [56] and recently VITT [16]. In this section, we address vaccine constituents and their involvement in the pathogenesis of VITT and the subsequent thrombo-inflammatory mechanisms.

It is pivotal to identify the vaccine component that is able to bind PF4, exposing a neoepitope and inducing the production of anti-PF4 antibodies from marginal zone B-cells, to be able to modify adenoviral vector-based vaccines in a way that VITT would be avoided. The first key insights into the ChAdOx1 vaccine and its association with VITT revealed that PF4 addition induces the aggregation of amorphous material in the ChAdOx1 vaccine and enables complex formation with the vaccine and anti-PF4 antibodies derived from VITT patients [16]. Interestingly, adenovirus-derived hexon proteins are identified to be one of the vaccine constituents, directly implicated in the formation of these ternary complexes. Baker et al. resolved the structure of the adenoviral hexon protein from the ChAdOx1 vaccine, predicted potential epitopes on PF4 by Brownian dynamics simulations and demonstrated through surface plasmon resonance that PF4 and the purified ChAdOx1 viral vector interact with each other [57]. Consistent with electrostatic surface calculations of ChAdOx1 hexon proteins, detailed analysis from DLS experiments showed that the association between PF4 and unassembled ChAdOx1 hexons is a charge-driven process [45]. Furthermore, a slight increase reported in the size of purified virion particles after addition of PF4, as demonstrated by DLS [45], is in congruence with the study by Baker et al. [57]. However, complexes between PF4 and purified ChAdOx1 virions could not be demonstrated by microscopy imaging [45], attributed to the presumably weak association between the two with a dissociation rate (K_d_) of 300 nmol. Rather than showing complexes between PF4 and the adenovirus itself, electron microscopy shows complexes between PF4 and amorphous material in the vaccine. Also, DLS experiments indicate that PF4 forms complexes with the virus-free supernatant from the ChAdOx1 vaccine and not with the virus particles [45]. These observations suggest the presence of vaccine components besides the intact virus/virion particles, which might be instrumental in neoepitope formation on PF4.

As sulfated glycosaminoglycans (GAGs) are acknowledged to be one of the different polyanions triggering HIT [58], they were previously considered in causing VITT as well, but due to the different reactivity to PF4-heparin complexes in HIT and VITT their involvement in VITT was shown to be unlikely. Furthermore, the concentration of GAGs in the ChAdOx1 vaccine was below the detection limit [59]. Nuclear magnetic resonance (NMR) investigation to identify other small molecules exposed the presence of 100 μM ethylene diamine tetra acetate (EDTA), as a constituent of the ChAdOx1 vaccine [16]. Being a polyanion, EDTA was a presumed candidate to interact with positively charged PF4. However, the role of EDTA in neoepitope generation and complex mediation between PF4 and anti-PF4 antibodies was already excluded by competitive assays with Ferric chloride [46]. However, as VITT has an ultra-rare occurrence, it might be orchestrated by a whole series of pathogenic changes with EDTA exerting another bystander role. EDTA is most likely relevant for inducing the inflammatory milieu needed for the anti-PF4 response. By binding Ca^2+^, EDTA disrupts the integrity of E-cadherin, an essential component of endothelial cell junctions. By disrupting endothelial cell junctions, EDTA in the ChAdOx1 vaccine locally increases vascular permeability in mice [16] and zebrafish [45] and presumably also in humans [16], which allows vaccine components to enter the circulation, thereby causing inflammation, e.g., by binding of non-assembled virus proteins to toll-like receptors (TLRs).

EDTA might be an example for different (and so far, unknown) factors that could lower the threshold for activation of the B-cells in VITT analogous to HIT [60]. Observations that T-REx HEK293 cell line-derived proteins from ChAdOx1 are recognized by endogenous IgG antibodies, as demonstrated by immunoblotting experiments [46], strengthen the hypothesis that host cell-derived proteins support pro-inflammatory reactions after vaccination, potentially accentuated by EDTA-induced capillary leakage. Capillary leakage induced by EDTA is probably also relevant for transferring vaccine–PF4 complexes to immune cells in the spleen.

Mass spectrometry and proteomic analysis showed that ChAdOx1 harbors a high concentration of proteins derived from T-REx HEK 293, a cell line used to propagate the virus vector [16,45]. The protein concentration derived from host cells is considerably higher in ChAdOx1 (19.1–33.8 μg per vaccination dose) than in Ad26-COV-2S (0.04–0.19 μg per vaccination dose) [45]. However, the proteins identified in both ChAdOx1 and Ad26-COV-2S are largely different with no overlap [45]. This makes it even more challenging to pin down potential proteins that elicit neoepitope formation on PF4. Probably, the cell line-derived proteins have only a bystander role in the pathogenesis of VITT by triggering an inflammatory response. Altogether, in comparison to ChAdOx1, Ad26-COV-2S vaccine has lower host cell-derived proteins and free hexon particles, and lack EDTA-induced vascular permeability. This might explain the lower incidence of VITT after vaccination with Ad26-COV-2S [61]. Recently several patients have been identified who presented with a VITT-like syndrome, including thrombocytopenia, thrombosis at unusual sites, high D-dimer levels and PF4-dependent, platelet-activating antibodies [62]. Several of these patients had a virus infection preceding the VITT-like complications. The only obvious common factor in VITT patients after Ad26-CoV2S or ChAdOx1 vaccination and VITT-like syndromes after viral infections is the virus. This hints towards a causative role of the virus in the pathogenesis of VITT [62].

## 5. Possible Events after Vaccination

Possible pathological mechanisms involved in VITT after vaccination are discussed in this section. Within the first two days after vaccination, vaccine constituents prompt neoepitope generation on PF4. A pro-inflammatory milieu is formed as a result of the vaccine components released into the vasculature, either by accidental intravascular injection, or EDTA mediated vascular leakage. In addition, preformed natural IgG antibodies detect vaccine contaminants, such as host cell-line proteins, as antigens and form complexes with them. These natural IgG antibodies normally have a role in “clearing” degraded cell proteins. These immune complexes augment the pro-inflammatory milieu. Within the vasculature, the unassembled viral hexon molecules and intact virions interact with PF4 and platelets [16] (Figure 5a). The resulting presumed neoepitope generation on PF4 and the pro-inflammatory milieu act together to trigger anti-PF4 antibody production by B-cells, after expansion and isotype switching of one or a few B-cell clones [63,64] (Figure 5b). These complexes may also boost an immune response against PF4–hexon complexes developed after previous viral infections [65,66]. From five to thirty days post vaccination, high titers of anti-PF4 IgG antibodies are observed in the circulation, followed by thrombosis and thrombocytopenia [8]. While the vaccine components that induce the production of anti-PF4 antibodies are still unresolved, the mechanism of how thrombo-inflammation is mediated by pathological anti-PF4 antibodies in VITT via FcγIIa receptor-dependent cell activation is well derived from HIT [16]. Both the VITT patient serum and purified anti-PF4 antibodies activate neutrophils in the presence of platelets and induce neutrophil extracellular traps (NETs), a process known as NETosis [16]. In vivo NETosis is further corroborated by the presence of NET biomarkers such as cell-free DNA, citrullinated histones and myeloperoxidase (MPO) in the serum of VITT patients, and neutrophil Elastase (NE) and MPO in platelet-rich cerebral sinus vein thrombi [16]. The anti-PF4 antibodies in VITT directly activate monocytes, which form extracellular traps (ETs) [67], although to a lesser extent compared to neutrophils. Furthermore, also platelet–monocyte aggregates are identified in VITT patients [68,69], which imply the activation of both platelets and monocytes. Earlier studies show that platelet–monocyte aggregates, mediated by P-selectin and integrins, are essential for the expression of tissue factor (TF) by monocytes [70,71]. Similar to this mechanism, the platelet–monocyte aggregates found in VITT could as well contribute to the production of TF, leading to subsequent thrombosis. Concomitantly, the complement pathway is activated in VITT patients [72] and autopsy results [73] reveal the detection of complement pathway components, C1r and C4d, on the vascular endothelial cell surface. This indicates that endothelial cells are activated by the complement pathway in VITT, which could result in the expression of TF and von Willebrand factor (vWF). It is known in HIT that vWF forms complexes with PF4, to which HIT antibodies bind [74] and then activate neutrophils via their Fc part, exacerbating thrombosis. A similar mechanism could entail in VITT to propagate thrombus formation (Figure 5c, upper panel).

Although concerns about the cross-reactivity of PF4 and vaccination-induced translation of anti-spike protein antibodies had been raised [76,77], it is unlikely that the SARS-CoV-2 spike protein itself induces VITT [78]. Evidence for the independence of immune responses to PF4 and the SARS-CoV-2 spike protein lie in the lack of increase in anti-PF4 antibodies and potential subsequent thrombocytopenia or thrombosis in former VITT patients who contracted COVID-19 later [79]. Another theory including cross-reactions with the spike protein underlines the possibility of random splicing events leading to transcription of abnormal forms of the spike protein, e.g., leading to the formation of immunological active soluble spike protein [77] able to induce a vascular inflammatory response and thrombosis. However, these aberrant splicing activities seem to be very rare [80]. While it is very unlikely that they are the main cause of VITT, it cannot be excluded that these events contribute to endothelial cell inflammation and thereby increase the risk for thrombosis. VITT is so rare that it is possible that only a combination of events leads finally to the breakthrough of clinical symptoms.

## 6. The Antibody Landscape within and beyond VITT

VITT patients develop anti-PF4 antibodies that are non-pathogenic [81] and antibodies that are pathogenic, i.e., able to activate platelets [3]. PF4-dependent platelet-activating antibodies in VITT gradually decrease over a median range of 15.5 weeks, after VITT onset in most patients [82]. Anti-PF4 antibodies as detected by anti-PF4-heparin IgG ELISA last much longer than pathogenic, platelet-activating anti-PF4 antibodies [83]. Besides a simple decrease in antibody titer, potentially also the functionality of VITT anti-PF4 antibodies changes over time. Nazy et al. provided preliminary evidence that VITT patient sera causing strong platelet activation without the need of PF4 addition typically contain two types of antibodies, one group binding to the VITT epitope and one group binding to the HIT epitope [84]. It might be that different combinations of antibodies have slightly different activation capacities.

FcγIIa receptors are activated when cross-linked by immune complexes. Monomeric IgG cannot activate the receptor. Immune complexes occur in HIT, because heparin is cross-linking PF4, leading to macromolecular complexes. Even if a cofactor in the vaccine might cross-link PF4, it is extremely unlikely that this cofactor is still present in the circulation five to twenty days after vaccination, when VITT clinically manifests. In the case of VITT, the anti-PF4 antibodies themselves cross-link PF4 and form large multi-molecular complexes. Therefore, an optimal stoichiometric ratio of both antigen and antibodies is critical for immune complex formation and platelet activation [85]. The rationale behind this observation lies in the fact that the presence of a higher proportion of either PF4 or VITT antibodies would be thermodynamically unfavorable to form adequate immune complexes, thereby limiting their potential to activate platelets [85]. This has relevance for in vitro tests for anti-PF4 antibodies. Functional tests for VITT antibodies become more sensitive when, in case of a negative test result of the functional assay, despite strong reactivity in the ELISA test, the serum was diluted 1:4–1:10. This reduces the concentration of anti-PF4 antibodies in sera with extremely high titers of these antibodies and facilitates a more favorable stoichiometric ratio of both binding partners for complex formation between the antibodies and PF4, with consecutive platelet activation [85].

Antibodies are glycoproteins which harbor N-linked oligosaccharides at Asparagine 297 (Asn-297) on their heavy chains. This glycosylation is essential for the binding of antibodies to Fc receptors on various effector cells [86]. Varying post-translational glycosylation at Asn-297, along with subclass switching, is a key immune mechanism to generate a diverse effector repertoire of antibodies [86]. Aberrant glycosylation patterns of antibodies is manifested in different autoimmune and infectious conditions [87,88]. Changes in the glycosylation pattern over time might be a reason for the much faster decline of platelet activating, pathogenic anti-PF4 antibodies in comparison to antibodies which are detectable by ELISA.

Modification of the glycosylation site of VITT antibodies might also have therapeutic implications. A study by Vayne et al. demonstrated that a deglycosylated VITT-like monoclonal antibody is unable to activate platelets via the FcγIIa receptor [89]. The deglycosylated monoclonal antibody competitively inhibits the binding of pathogenic VITT antibodies to activate platelets and thrombus formation. Such deglycosylated monoclonal antibodies have the potential to be used as treatment in acute VITT, besides anticoagulants. The current way to block FcRIIa-dependent platelet activation is treatment with high dose intravenous immunoglobulins, which bind to the Fc receptors and inhibit binding to the PF4–antibody immune complexes.

Different parameters of anti-PF4 antibodies were further characterized, such as clonality and subtype. While HIT antibodies are polyclonal, VITT antibodies are either monoclonal or oligoclonal. In addition, most VITT antibodies are of the IgG1 subclass [63]. The antigen-binding Fab region of antibodies, formed by heavy and light chain variable regions, was evaluated in different VITT patients, yielding surprising results. In all patients, the genetic source of the light chain variable region was the same gene (IGLV3-21*02), producing a transcript that codes a highly conserved acidic amino acid (i.e., negative charge) motif [64]. In addition, Huynh et al. identified that VITT antibodies bind to a Lysine and Arginine-rich (i.e., positively charged) amino acid moiety on the PF4 surface [25]. The binding sites and binding characteristics of anti-PF4 antibodies induced after ChAdOx1 or Ad26-COV-2S vaccines do not differ. This had been determined by Alanine scanning mutagenesis, and by biolayer interferometry [25,90]. These results emphasize charge-based interactions between PF4 and anti-PF4 VITT antibodies besides typical structure-based interactions between antibodies and antigens. Despite unveiling crucial structural and functional aspects of antibodies in VITT, these studies are limited to a very small sample size, underlining the need to involve larger patient cohorts or suitable in vivo animal models in a comprehensive study to further characterize anti-PF4 antibodies involved in VITT.

Anti-PF4 antibodies might not be the only cause for thrombocytopenia in VITT. About 30% of VITT patients, besides having anti-PF4 antibodies, also possess antiplatelet antibodies directed against platelet glycoproteins. This suggests that, like in immune thrombocytopenia (ITP), antiplatelet antibody-mediated opsonization of platelets could further aggravate thrombocytopenia (Figure 5c, lower panel) and increase the risk for bleeding [75].

In view of the above reports, VITT patients have a diverse repertoire of antibodies (platelet-activating and non-activating anti-PF4 antibodies, and antiplatelet antibodies) that may even act synergistically to manifest thrombosis and thrombocytopenia. Conversely, VITT-like antibodies could be detected beyond VITT, as in a patient with monoclonal gammopathy [12]. An IgG-κ type monoclonal paraprotein, which binds to PF4 and activates platelets via their FcγIIa receptor, has been identified in this patient who presented with recurrent deep vein thrombosis, pulmonary embolism, stroke and thrombocytopenia. The monoclonal paraprotein of the patient has a striking similarity with PF4-dependent platelet-activating anti-PF4 antibodies of VITT [12]. Such patients are probably more frequent [91], and anti-PF4 antibodies could be one reason for the increased rate of thrombosis in patients with monoclonal gammopathy. An increasing number of patients are currently identified with platelet-activating anti-PF4 antibodies causing thrombocytopenia and thrombosis, but without previous heparin exposure or vaccination. Some of them had a preceding adenoviral infection [13]. New assays allow to identify VITT-like anti-PF4 antibodies in these patients, which are related to severe immunothrombosis independent of heparin or (COVID-19) vaccination. Potentially, such patients remained previously undiagnosed due to negative results in rapid HIT-tests and heparin-dependent functional tests. This indicates that VITT-like anti-PF4 antibodies are probably a cause of thrombosis not recognized in patients so far [33,62].

## 7. Mechanistic Insights from Murine Models of VITT

VITT as an adverse event of COVID-19 vaccination is affecting only a small group of patients, limiting the availability of patient samples for mechanistic studies [89]. Although some promising in vitro models using vWF- or collagen-coated microchannel flow chambers mimicking venous blood flow were established [26,67], in vivo models are better suited to study the clinically relevant systemic impacts of VITT [67]. Furthermore, with more patients currently being recognized with VITT-like disorders, the use of preclinical models might help to elucidate downstream pathomechanisms or therapeutic options in VITT or VITT-like disease.

The first work to unravel VITT pathogenesis as a two-step model used both patient-derived data and insights from a murine model [16]. The model used in this study was the Miles edema model [92] challenged with intradermal injections of the ChAdOx1 vaccine to observe the ability of vaccine components to induce leaky vessels as a hallmark of inflammation [16]. Indeed, this experiment unraveled the extra vasal distribution of vaccine components and pointed towards the potential ability of vaccine components, like EDTA, to promote vascular leakage [16].

Nicolai et al. addressed mechanisms of VITT using a murine immunization model. They applied different regimes of vaccination with ChAdOx1 in C57Bl6 mice [75]. The group found convincing insights regarding the mechanisms, by which vaccination with an adenovirus vector vaccine induces anti-platelet antibodies. Platelets bind adenovirus by the Cocksackie/Adenovirus receptor (CAR) [93], which is expressed on mice and human platelets alike [94]. Intravenous vaccine injection leads to the binding of adenovirus to platelets and platelet activation. These virus particle-coated platelets are then transported to the spleen by the blood flow and accumulate in the marginal zone. The co-presentation of virus antigens and platelet glycoproteins by platelet–adenovirus complexes then breaks tolerance and triggers an anti-platelet immune response [75]. Nicolai et al. furthermore suspected that this immune response subsequently leads to an increased splenic platelet clearance as an underlying mechanism of the observed thrombocytopenia, depending on the epitope and antibody titer [75]. This mechanism is well established for ITP. However, it is unresolved whether this model is also suitable to explain immunization against PF4 complexes. Probably PF4 is also presented to the immune cells on the surface of platelets.

Due to the similarities between HIT and VITT, one important finding from HIT models suggests that the production of subsequent IgG antibodies relies on specialized marginal zone B-cells [95]. Potentially, the immune response to PF4 is part of the inborn antibody repertoire, especially for anti-PF4 IgM antibodies. In HIT even PF4 knockout mice (that had previously never been exposed to PF4) produced anti-PF4 antibodies when challenged with polymicrobial sepsis [96].

For systematic studies on the mechanisms of thrombo-inflammation and thrombosis in VITT, mouse models would be ideal. However, mice lack the equivalent of the human FcγIIA receptor on platelets [97]. To better understand the role of FcγIIA receptors in vivo, transgenic mice expressing the human FcγIIA receptor (hFcγRIIA) were generated more than two decades ago [97]. In combination with a transgene of human PF4 (hPF4), the hFcγRIIa+/hPF4+ double transgenic mouse model was generated initially to study HIT after heparin administration [98,99,100]. In the meantime, Leung and colleagues established the hFcγRIIa+/hPF4+ double transgenic mouse with the intravenous injection of purified VITT IgG as an in vivo VITT model [67], with evidence that immune complexes formed by VITT IgG and PF4 induce clot formation in vivo [67]. Based on first insights from patients on the importance of platelet involvement in NETosis [16,101], the transgene VITT model shows that NETosis directly drives thrombosis in VITT [67]. In addition, it was possible to point out that neutrophils could be directly activated by VITT IgG and PF4 even when platelets were not involved [67], indicating a general involvement of the innate immune system in VITT.

Although clinically therapeutic dose anticoagulation and high-dose intravenous immunoglobulins seem to be highly effective in VITT [6], more insights are needed, especially for patients who have persisting platelet activating anti-PF4 antibodies. Murine models will be able to contribute to such studies. Therefore, main findings regarding VITT available from these models are summarized in Table 1. Even if these models might not be able to clarify understanding of the very rare occurrence of VITT following adenovirus-vector vaccine injection, based on the presented research they might enable to study the pathological downstream mechanisms and might even help to improve future therapy. One interesting question is whether the inhibition of Bruton Tyrosin Kinase (BTK) in platelets and leukocytes might be able to prevent VITT antibody-related thrombotic complications [102]. From the perspective of vaccine developers, better understanding of the detailed mechanisms of the immune reactions to adenoviral vector-based vaccines is highly desirable. This includes defining what triggers a first anti-PF4 immune response as a primer for B-cell activation during the secondary immune response after vaccination. This may help to produce vaccines with a lower risk for VITT. Furthermore, understanding the underlying mechanisms of VITT may help to identify useful biomarkers for an enhanced risk of developing VITT, although, due to the rarity of this adverse vaccination reaction, biomarkers probably will never be applied in the context of a large vaccination campaign.

## 8. Conclusion and Further Perspectives

In addition to providing abundant insights into VITT pathogenesis generated from clinical observations and ex vivo experiments, murine in vivo models of VITT are on the rise to clarify basic mechanisms and test the refinement of therapeutic approaches. The use of these models together with monoclonal VITT-mimicking antibodies is foremost important, as availability of patient samples for mechanistic studies is very limited and ex vivo test systems often lack insights regarding systemically relevant mechanisms. Although research has generated intriguing insights into the binding site of VITT anti-PF4 antibodies and the structure of their hypervariable region antigen binding site, the exact trigger(s) which mediates the secondary immune response in VITT and the corresponding primary immunization agent(s) are still elusive.

The usually applied in vitro approaches to identify suitable binding partners of PF4 that are able to cause potential conformational changes hereby inducing the immune reaction to PF4 would be straightforward. However, the very high affinity of anti-PF4 antibodies in VITT, and the phenomenon of epitope spreading [42] still cause major challenges. Also, the role of T-cells and the subpopulation of involved B-cells are unknown. Animal models will help to clarify the mechanistic orchestration of underlying immune mechanisms.

Furthermore, very recent preliminary data [103] indicate that among patients with clinically defined TTS, thrombocytopenia and (recurrent) thrombotic events, anti-platelet antibodies can be detected. These antibodies induce procoagulatory platelets. This indicates a second mechanism causing TTS that differs from the anti-PF4 antibody mediated mechanism typically seen in VITT [12]. This further underscores the importance of detailed studies on the mechanisms of VITT and presumably other subforms of TTS to refine diagnosis and treatment in these patients, and to develop rational approaches to increase vaccine safety.

Finally, VITT-like syndromes and VITT-like anti-PF4 antibodies have been identified retrospectively in patients who presented before 2020 [62]. This strongly indicates that VITT/VITT-like disorders existed before and had not been recognized. Although common overlapping mechanisms still have to be examined, a possible common mechanism between VITT after vaccination with ChAdOx1 or Ad26-CoV-2S and VITT-like syndromes could lie in the pre-exposure to viral particles, e.g., during previous viral infections (Figure 6). How frequently VITT-like antibodies are the underlying cause in patients presenting with thrombocytopenia, thrombosis at unusual sites and high D-dimer levels, needs now to be assessed in larger patient cohorts.

## Figures and Tables

**Figure 1 jcm-12-06126-f001:**
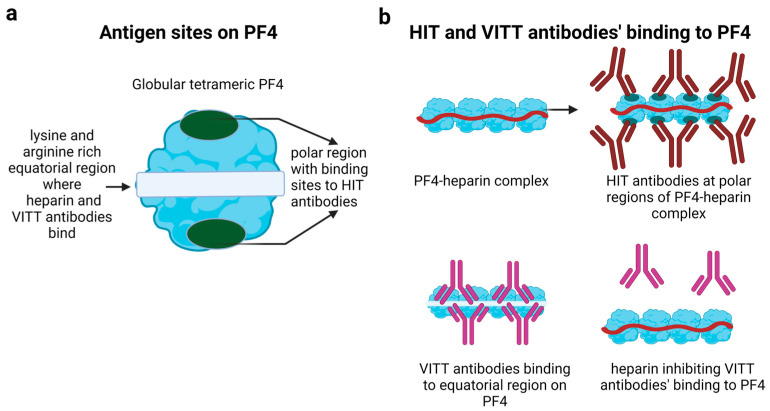
Platelet factor 4 (PF4) antigen binding. (**a**) PF4 in its tetrameric form has two major binding sites. The first consists of the polar regions able to bind heparin-induced thrombocytopenia (HIT) antibodies. The second one is located in the lysine- and arginine-rich equatorial plane. Heparin as well as vaccine-induced immune thrombotic thrombocytopenia (VITT) antibodies are able to bind at this site. (**b**) Heparin binds to the equatorial plane on PF4, causing a conformational change and exposing the polar regions, recognized by HIT antibodies (**upper panel**). VITT antibodies bind to the equatorial plane of PF4 molecules. Binding of heparin can lead to a competitive inhibition of VITT antibody binding to the equatorial plane (**lower panel**).

**Figure 2 jcm-12-06126-f002:**
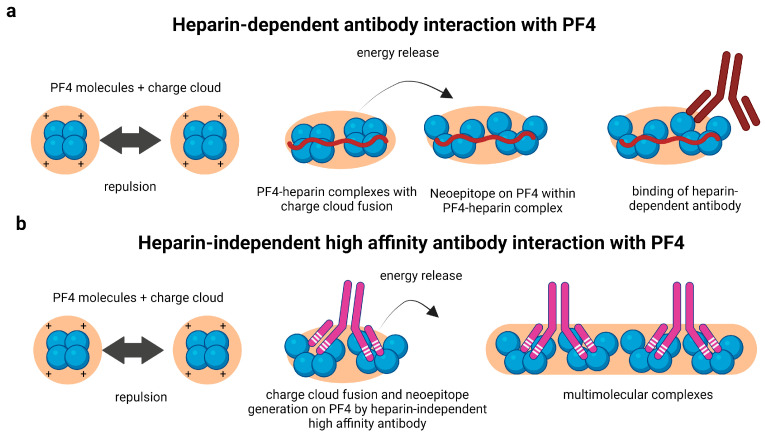
Interaction of PF4 with heparin-dependent antibodies and heparin-independent autoantibodies. (**a**) PF4 tetramers are surrounded by a positive charge cloud. Two PF4 tetramers experience strong repulsive forces due to the positive charge cloud surrounding them (**left panel**). Charge-based interaction of heparin with PF4 enables the fusion of positive charge cloud around the molecules inducing energy release and neoepitope generation (**middle panel**). Heparin-dependent antibodies recognize and bind to the generated neoepitope on PF4 (**right panel**). (**b**) PF4 molecules are surrounded by positive charge cloud (**left panel**). High affinity heparin-independent autoantibodies induce conformational changes and generate neoepitope on PF4 molecules (**middle panel**). The energy released is further utilized to generate multimolecular complexes (**right panel**).

**Figure 3 jcm-12-06126-f003:**
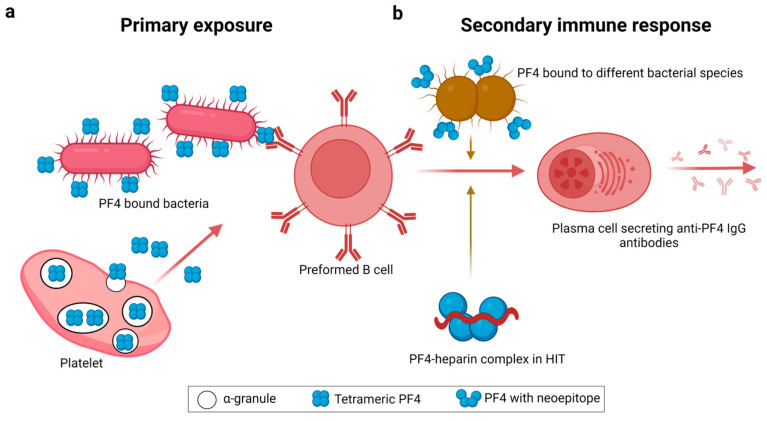
A theory to explain the role of PF4 in the manifestation of rapid secondary immune response. (**a**) During the primary exposure, PF4 secreted by platelets interacts with a bacterial species and the PF4 neoepitopes on bacterial complexes are recognized by preformed B-cells. (**b**) After the interaction of a second bacterial species with PF4, the same neoepitope is generated on PF4 (**upper panel**), inducing the differentiation of preformed B-cells to plasma cells which secrete IgG antibodies into the blood stream (**middle panel**). In heparin-induced thrombocytopenia (HIT), the neoepitope is generated in PF4 after interaction with heparin (**lower panel**). This triggers pre-formed B-cells to differentiate into plasma cells, which secrete high titer IgG antibodies. VITT is probably caused by a similar mechanism which is still not well understood.

**Figure 4 jcm-12-06126-f004:**
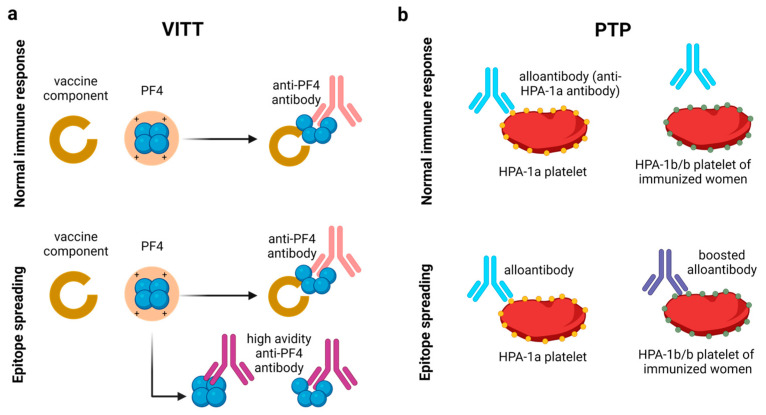
Epitope spreading in VITT and post-transfusion purpura (PTP). (**a**) Normal immune response in VITT entails an interaction of PF4 with vaccine component resulting in conformational change in PF4 and the anti-PF4 antibody binds to vaccine–PF4 complex (**upper panel**). A boosted immune response possibly causes epitope spreading, due to which anti-PF4 antibodies are produced which can bind with high avidity to PF4 without the presence of any vaccine cofactor (**lower panel**). (**b**) In post-transfusion purpura, in normal immune response, the anti-HPA1a alloantibodies only bind to HPA1a platelets, but not the HPA1b platelets of the host (**upper panel**). A boosted immune response by epitope spreading results in the production of anti-HPA1a antibodies binding to both HPA1a platelets as well as the HPA1b platelets of the host (**lower panel**).

**Figure 5 jcm-12-06126-f005:**
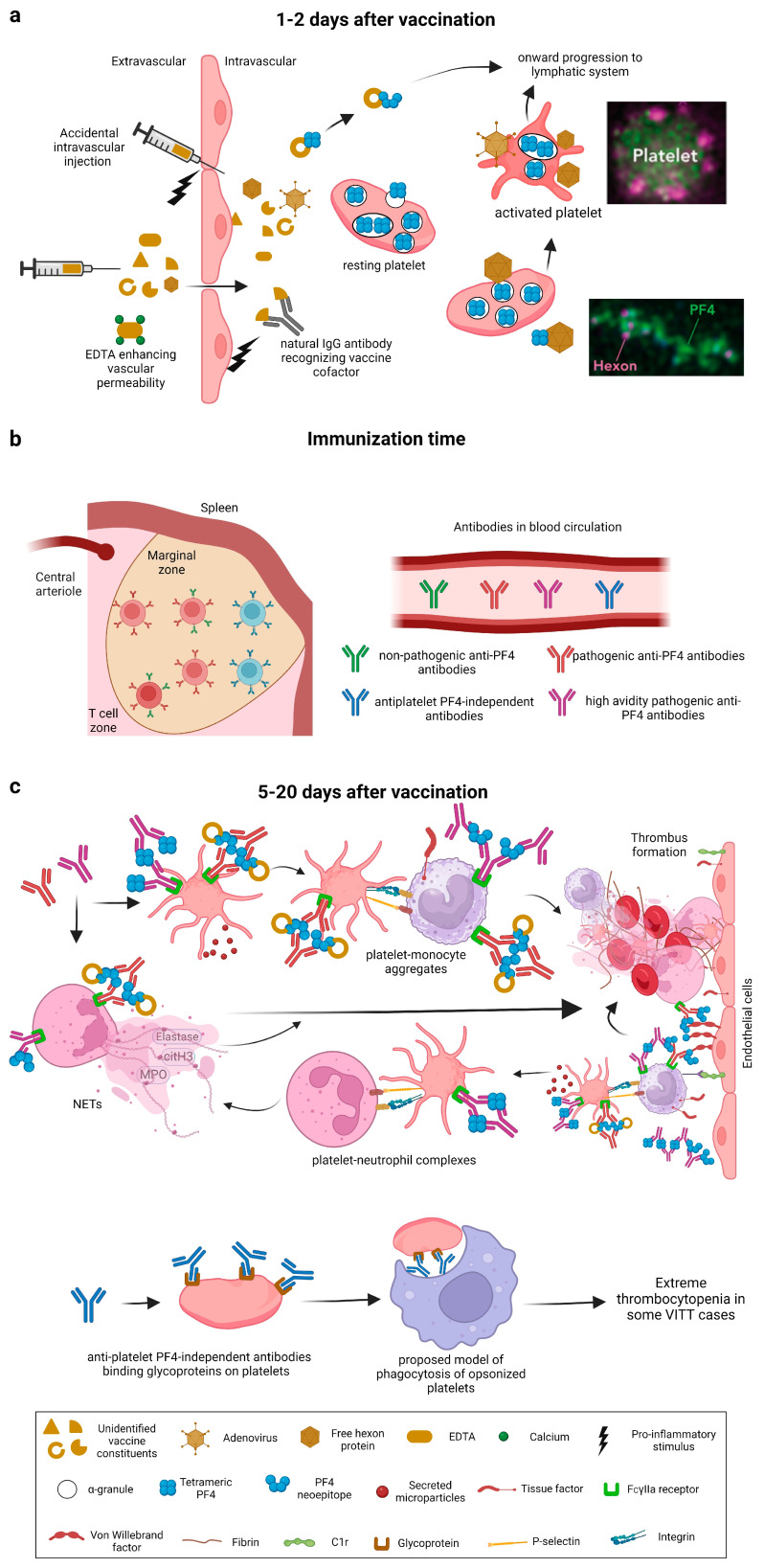
Progression of VITT: current evidence and proposed models. (**a**) After vaccination, vaccine components can enter the intravascular space via two routes. The first route is by accidental intravascular administration. The second route is mediated by the presence of EDTA in the vaccine composition that chelates calcium from cell–cell junctions and acts as proinflammatory trigger by enhancing vascular permeabilization. Within the first two days after vaccination with adenoviral vectors, neoepitope is generated on PF4 after interaction with viral hexons or so far unidentified vaccine constituent. The photomicrographs depicting the interaction of viral hexons with platelets and PF4 are based on [16]. (**b**) Within immunization time, marginal zone B-cells recognize neoepitope on PF4 and secrete non-pathogenic as well as pathogenic anti-PF4 antibodies, high avidity pathogenic anti-PF4 antibodies and anti-platelet PF4-independent antibodies carried into the bloodstream. (**c**) Within 5–20 days after vaccination, thrombosis and thrombocytopenia are manifested. Pathogenic anti-PF4 antibodies as well as high avidity pathogenic anti-PF4 antibodies result in FcγIIa receptor-mediated platelet activation, NETosis and to a minor extent also extracellular trap formation by monocytes, consequently enabling thrombus formation. Thrombus propagation is additionally favored by interactions of PF4 with von Willebrand Factor and monocytes with complement component 1r (C1r) expressed on endothelial cells. Monocyte–platelet aggregates accentuate thrombosis by expression of tissue factor (TF) (**upper panel**). Anti-platelet PF4-independent antibodies bind to glycoproteins on platelets, and a hypothesis was put forth [75] that this binding facilitates opsonization of platelets and subsequent clearing by phagocytosis, resulting in exacerbated thrombocytopenia in some cases of VITT (**lower panel**).

**Figure 6 jcm-12-06126-f006:**
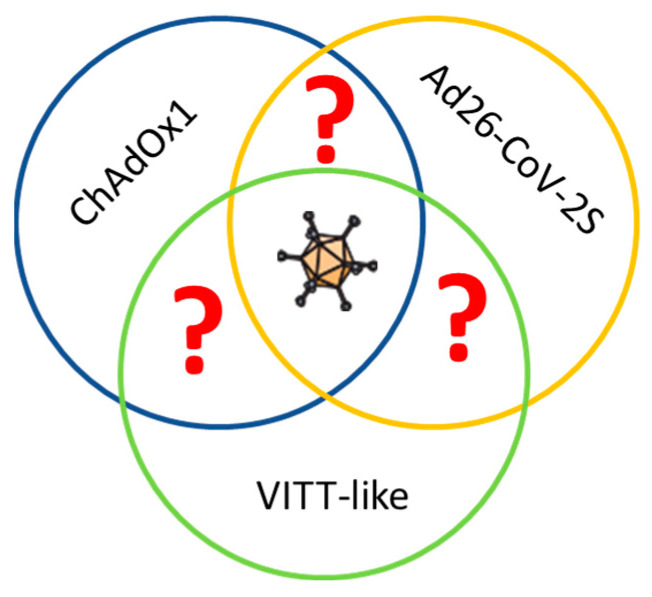
Contact to viral particles is common in VITT after vaccination and VITT-like disorders. Although common overlapping mechanisms still have to be examined, a possible common mechanism between VITT after vaccination with ChAdOx1 or Ad26-CoV-2S and VITT-like disorders could lie in the pre-exposure to viral particles, e.g., during previous viral infections.

**Table 1 jcm-12-06126-t001:** Summary of the main results on VITT-related research in murine models.

Cited Work	Used Model	Core Statements
Greinacher, A. et al. Blood 2021 [16]	Miles edema model [92]	ChAdOx1 nCov-19 vaccine components are able to induce leaky vessels as a hallmark of inflammation. This pro-inflammatory milieu can facilitate VITT development.
Nicolai, L. et al. Blood 2022 [75]	ChAdOx1 immunization model in C57Bl6 mice [75]	Intravenous injection of ChAdOx1 nCov-19 triggers the formation of platelet-adenovirus aggregates and platelet activation. The immune response subsequently leads to increased splenic platelet clearance as an underlying mechanism of the observed thrombocytopenia.
Krauel, K. et al. Thromb Haemost 2016 [96]	PF4-knockout mice [96]	The immune response to PF4 is part of the inborn antibody repertoire.
Leung, H.H.L. et al. Nat Commun 2022 [67]	hFcγRIIa+/hPF4+ double transgenic mice [98] immunized with VITT IgG	VITT antibodies directly stimulate neutrophils to release NETs. VITT antibodies activate blood cells via FcyRII receptor. VITT antibody-induced thrombi contain platelets, neutrophils, fibrin, extracellular DNA and citrullinated histone H3.

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
