# Peer review of "Vaccine-Induced Immune Thrombocytopenia and Thrombosis (VITT)—Insights from Clinical Cases, In Vitro Studies and Murine Models"

_jcm, 2023, doi:10.3390/jcm12196126_

Round 1
Reviewer 1 Report
Personally, I am in favor of discussing new insights and possible mechanisms of action. The article is therefore food for thought for me.
However, the article has been written as a grant application, with the focus on hypotheses for which evidence has largely yet to be found. It is difficult for the uninitiated reader now to distinguish between proven and speculated. I therefore think it would be good to put more emphasis (especially with the figures) on the fact that they are hypothetical working mechanisms.
more specific comments
The distinction between TTS and VITT is not yet entirely clear to me. If VITT cases are defined as patients with strong PF4 antibodies, what is the underlying cause for TTS? We did also see many cases in which patients develop a thrombosis post vaccination (with thrombocytopenia), but that may just as well be a coincidence and not vaccination related. Especially after it became known that VITT can occur after vaccination, clinicians requested very low-threshold VITT diagnostics.The comparison between PTP and VITT is wonderful, but to have to use this to explain VITT immediately shows the weak basis for the overall theory.
The part about heparin use can cause misunderstanding for treating physicians and should be written with more care.
The comparison between PTP and VITT is wonderful, but to have to use this to explain VITT immediately shows the weak basis for the overall theory.
The claim that VITT-like anti-PF4 antibodies are a cause for thrombosis, again is very theoretical and needs at least more investigation to use the word 'indicate'.
Author Response
Reviewer 1:
Comments and Suggestions for Authors
Personally, I am in favor of discussing new insights and possible mechanisms of action. The article is therefore food for thought for me.
However, the article has been written as a grant application, with the focus on hypotheses for which evidence has largely yet to be found. It is difficult for the uninitiated reader now to distinguish between proven and speculated. I therefore think it would be good to put more emphasis (especially with the figures) on the fact that they are hypothetical working mechanisms.
Answer:
We thank the reviewer for his/her favourable opinion on our article. As per the reviewer’s suggestion, we have more explicitly stated in the figure legends and in the text where hypothetical mechanisms are discussed. Please see the marked changes throughout the whole text.
More specific comments
- The distinction between TTS and VITT is not yet entirely clear to me. If VITT cases are defined as patients with strong PF4 antibodies, what is the underlying cause for TTS? We did also see many cases in which patients develop a thrombosis post vaccination (with thrombocytopenia), but that may just as well be a coincidence and not vaccination related. Especially after it became known that VITT can occur after vaccination, clinicians requested very low-threshold VITT diagnostics.
Answer:
TTS is an overarching term to describe thrombosis and thrombocytopenia syndromes and VITT is a sub-group of TTS, where anti-PF4 antibodies are accepted as mediators of thrombosis and thrombocytopenia. TTS can be caused by a variety of factors. Most often thrombocytopenia and thrombosis are coincidence, e.g. in patients with cancer receiving vaccination. However, there seem to be some factors triggered by vaccination which cause TTS including platelet activating antibodies towards platelet glycoproteins (Thiele, T.; Esefeld, M.; Handtke, S.; Rath, J.; Schönborn, L.; Antovich, J.; Lotfi, K.; Holmström, M.; Harasser, L.; Feistritzer, C.; et al. Thrombotic Thrombocytopenia Syndrome after Vaccination against COVID-19 Associated with Antiplatelet Antibodies. ISTH Congress Abstracts 2023). Most interestingly, VITT antibodies seem to be present since many years but have not been recognized. Probably they are triggered by viral infections. A recent study shows that a preceding infection with adenovirus could subsequently contribute to the manifestation of VITT-like symptoms (Warkentin, Theodore E et al. “Adenovirus-Associated Thrombocytopenia, Thrombosis, and VITT-like Antibodies.” The New England journal of medicine vol. 389,6 (2023): 574-577. doi:10.1056/NEJMc2307721). This emphasizes the necessity of a detailed investigation into the possible interplay of preceding viral infections and the vaccine constituents to the onset of thrombosis and thrombocytopenia after vaccine administration.
- The part about heparin use can cause misunderstanding for treating physicians and should be written with more care.
Answer:
The WHO guidelines referenced in the article suggest treating the patients who develop thrombosis and thrombocytopenia post vaccination with non-heparin anticoagulants (NHACs), intravenous immunoglobulin (IVIG), steroids or heparin, in the absence of the availability of NHACs. The WHO report also shows the comparison of heparin and other anti-coagulants as therapeutic agents and conclude that during the usage of heparin, the harms are trivial, while the benefits are uncertain to large. Accordingly, we included this information in the current article (page.3, lines 102-119).
As for the sentence indicating 5% of VITT patients having anti-PF4 antibodies, we modified it (page.3, lines 109-111), also as per second reviewer’s suggestion, to limit any confusion.
- The comparison between PTP and VITT is wonderful, but to have to use this to explain VITT immediately shows the weak basis for the overall theory.
Answer:
It is our intention to introduce the readers with the concept of epitope spreading in VITT and to show occurrence of the phenomenon in other diseases. Furthermore, the comparison brings about the similarities of the antibodies, in both PTP and VITT. PTP is a suitable example where alloantibodies show autoantibody reactivity transiently after epitope spreading. After a few weeks, the antibodies no longer bind HPA-1 negative platelets.
Similarly, anti-PF4 antibodies are most likely generated as alloantibodies (against PF4 conformationally changed by vaccine cofactors), but after epitope spreading, they acquire the high-avidity binding capability to native PF4. These antibodies are also transient, although they persist longer than the autoreactive anti-GPIIbIIIa antibodies in PTP.
A detailed explanation of VITT is given in the subsequent sections.
- The claim that VITT-like anti-PF4 antibodies are a cause for thrombosis, again is very theoretical and needs at least more investigation to use the word 'indicate'.
Answer:
Agreeing to the viewpoint of the reviewer, we changed the sentence accordingly (page.14, lines 523-524).
Reviewer 2 Report
Global comments:
1. This is an excellent review, well-written, and very comprehensive, although some statements are discussable or are only working hypothesis. Concerning the PF4 reactivity of HIT and VITT antibodies, the authors report that the PF4 molecule is modified in both cases. Actually, available studies show that if in HIT the neoepitopes generated through the PF4 binding to heparin is the first trigger, some of the antibodies are also targeted to PF4 alone in some patients, even in the absence of heparin, and this was shown in the first reports , starting from 1992 (ref 47).
2. Various studies have shown that all patients with PF4 dependent HIT have antibodies that react with heparin and PF4 (HPF4) complexes, but a subset has antibodies reacting with PF4 alone. Reactivity to HPF4 results from exposition of neoepitopes following PF4 binding to heparin, whilst reactivity to PF4 could result from a prolonged or more intense exposure and/or immune response (targeted PF4 epitopes are then present on the intact protein, and antibodies generation could be the effect of the epitope spreading process).
3. In VITT, antibodies are targeted to the PF4 ring of positive charges, which is the same structure involved for the PF4 binding to heparin as shown by Huyn et Al (reference 25). Epitopes do not require any other component for being exposed. Antibody generation is then probably the result from the epitope spreading mechanism, when PF4 binds to “possible vaccine components”, like the adenovirus hexon protein. For this mechanism, binding to PF4 does not need to be that strong.
4. VITT has an ultra-rare occurrence, and resembles to some similar anti-PF4 antibodies generated in rare patients following viral infections, although the anti-PF4 titer and reactivity uses to be stronger. The understanding of this so rare occurrence remains a mystery, and the authors discuss cleverly the various possible hypothesis, including the possible presence of a pre-existing risk factor or a former immunization. This ultra-rare incidence excludes any general mechanism involving vaccine components interactions at the injection site, like the action of EDTA.
5. The authors cite many of their studies, with less references outside their working group, which is understandable from their very strong implication in that field. But some statements were reported before some of the references they cite (like anti-PF4 and anti-HPF4 antibody affinity links with platelet activation).
6. Lastly, the murine models presented are very interesting, but with only a weak usefulness for understanding the very rare occurrence of VITT following adenovirus-vector vaccines injection.
Concerning minor comments:
1. The authors should distinguish antibodies to heparin and PF4 complexes (HPF4) from antibodies to PF4, especially because those targeted to HPF4 require the PF4 binding to heparin or other polymeric polyanions (with a sufficient size to wrap around PF4 tetramer to modify its structure).
2. Page 3, line 94: the differential reactivity of anti-PF4 antibodies developed in VITT, depends on the assay used. It is rather usual with Elisas, and much less frequent in chemiluminescent assays. However, when the right assay conditions are met, (almost) all VITT antibodies react with HPF4, and not only 5%. Interestingly, anti-PF4 antibodies in VITT were identified by using HPF4 Elisas.
3. Page 3, lines 115-119: the presence of antibodies binding to HPF4, and a subgroup binding also binding to PF4 alone, was already reported since 1992 (reference 47), and many times after.
4. Page 4, lines 121-123: when the immune stimulus is a complex between PF4 and polyanions (big enough), antibodies developed are like those produced in HIT, even though some are reactive with PF4 alone (epitope spreading).
5. Page 4, lines 126-130: this feature was also shown in 2000 by extracting different anti HPF4/PF4 antibodies from the same patients with HIT, and separating them in groups of low and high affinity; only those with the highest affinity activated platelets (Amiral J, Pouplard C, Vissac AM, Walenga JM, Jeske W, Gruel Y. Affinity purification of heparin-dependent antibodies to platelet factor 4 developed in heparin-induced thrombocytopenia: biological characteristics and effects on platelet activation. Br J Haematol. 2000 May;109(2):336-41).
6. Page 6, lines 219-220: this reference (47) from 1992 already reported that some patients with HIT had antibodies binding to both HPF4 and PF4 (figure 1).
7. Page 7, lines 272-273: the very different reactivity of HIT and VITT antibodies to HPF4/PF4 excludes the complex formation with a polyanion.
8. Hexon is a candidate for binding PF4 and triggering the immune response, although its binding to PF4 is not that strong; but the immune response first targeted to the adenovirus hexon (or already preexisting from a viral infection) can in very rare cases be extended to PF4 complexed with it (epitope spreading). Lines 319-320 give a possible explanation for that occurrence.
9. Global presentation and discussion is very interesting and interestingly reported.
Author Response
Reviewer 2:
Global comments:
- This is an excellent review, well-written, and very comprehensive, although some statements are discussable or are only working hypothesis. Concerning the PF4 reactivity of HIT and VITT antibodies, the authors report that the PF4 molecule is modified in both cases. Actually, available studies show that if in HIT the neoepitopes generated through the PF4 binding to heparin is the first trigger, some of the antibodies are also targeted to PF4 alone in some patients, even in the absence of heparin, and this was shown in the first reports , starting from 1992 (ref 47).
Answer:
With regards to the comment above we emphasized on some aspects being working hypotheses at suitable passages throughout the whole text and figure legends. Please see the marked changes.
Furthermore, we tried to clarify the discrepancy of PF4 reactivity in HIT and VITT in section 2, describing in a bit more detail that there are indeed antibodies reacting to both, heparin-dependent (in the majority of cases) as well as heparin-independent in a subgroup of HIT patients. We also added suitable references as suggested (Amiral, J et al. 1992, doi:10.1055/S-0038-1656329/ID/JR_7/BIB ; Newman, P M and Chong, B H 1999, doi:10.1046/j.1365-2141.1999.01717.x ; Padmanabhan, A et al. 2015, doi:10.1182/blood-2014-06-580894 ). We currently work on a large study showing that at least 30% of HIT patients also have antibodies which react with PF4 alone. Please see according changes (page.3, lines 120-130)
- Various studies have shown that all patients with PF4 dependent HIT have antibodies that react with heparin and PF4 (HPF4) complexes, but a subset has antibodies reacting with PF4 alone. Reactivity to HPF4 results from exposition of neoepitopes following PF4 binding to heparin, whilst reactivity to PF4 could result from a prolonged or more intense exposure and/or immune response (targeted PF4 epitopes are then present on the intact protein, and antibodies generation could be the effect of the epitope spreading process).
Answer:
Part of this query was also answered in the paragraph regarding question 1. Furthermore, we tried to add an explanatory paragraph on that subject with focus on the hypothesis that reactivity in HIT to PF4 alone might be the results of prolonged exposure or intensified immune response mediated by effects of epitope spreading (page.3, lines 120-130, and page 4). This might also be the underlying mechanisms in some forms that are summarized with the term autoimmune HIT (lines 156-157).
- In VITT, antibodies are targeted to the PF4 ring of positive charges, which is the same structure involved for the PF4 binding to heparin as shown by Huyn et Al (reference 25). Epitopes do not require any other component for being exposed. Antibody generation is then probably the result from the epitope spreading mechanism, when PF4 binds to “possible vaccine components”, like the adenovirus hexon protein. For this mechanism, binding to PF4 does not need to be that strong.
Answer:
We fully agree with the reviewer. We changed the text to make the following points more clear to the reader: (i) heparin as well as heparin-independent PF4-antibodies in VITT have supposedly the same binding site on PF4 as shown by Huyn et al. and in the explanatory figure 1, (ii) due to heparin binding in this region a new epitope is exposed on the “pole” site of PF4. Whether PF4 is changed in its structure in the region of the heparin-binding site after complex formation of PF4 with unknown constituents of the vaccine is currently unknown. (iii) Induction of a strong immune response to PF4 without any conformational change in PF4 is also possible but in our opinion unlikely. Such “true” autoantibodies are typically long lasting and do not disappear within a few months. Please see changes (page.3, lines 88-100)
- VITT has an ultra-rare occurrence, and resembles to some similar anti-PF4 antibodies generated in rare patients following viral infections, although the anti-PF4 titer and reactivity uses to be stronger. The understanding of this so rare occurrence remains a mystery, and the authors discuss cleverly the various possible hypothesis, including the possible presence of a pre-existing risk factor or a former immunization. This ultra-rare incidence excludes any general mechanism involving vaccine components interactions at the injection site, like the action of EDTA.
Answer:
In the referred paragraph it was already stated that a role of EDTA in neoepitope generation could be excluded using competitive assays, however with the applied changes we tried to even make that point clearer. As VITT has an ultra-rare occurrence, it might be orchestrated by a whole series of pathogenic changes. EDTA is most likely relevant for inducing the inflammatory milieu needed for the anti-PF4 response. By its interaction with E-cadherin it allows vaccine components to enter the circulation, thereby causing inflammation, e.g. by binding of non-assembled virus proteins to TLRs. EDTA is just one factor that could lower the threshold for activation of the B-cells which finally produce VITT anti-PF4 antibodies. Please see the applied changes (page.8, lines 315-341).
- The authors cite many of their studies, with less references outside their working group, which is understandable from their very strong implication in that field. But some statements were reported before some of the references they cite (like anti-PF4 and anti-HPF4 antibody affinity links with platelet activation).
Answer:
We apologize and tried to include the work of others as much as possible. Please see the updated reference list.
- Lastly, the murine models presented are very interesting, but with only a weak usefulness for understanding the very rare occurrence of VITT following adenovirus-vector vaccines injection.
Answer:
We understand the concerns of the reviewer in this case. Nevertheless, we have the opinion that by reporting on VITT research in suitable mouse models, we may be able to provide the reader with new ideas for future attempts to elucidate downstream pathomechanisms or therapeutic options in VITT or VITT-like disease. In the first (page.14, lines 530-533) and last paragraph (page. 15, lines 580-583) of section 7 we tried to clarify this claim.
Concerning minor comments:
- The authors should distinguish antibodies to heparin and PF4 complexes (HPF4) from antibodies to PF4, especially because those targeted to HPF4 require the PF4 binding to heparin or other polymeric polyanions (with a sufficient size to wrap around PF4 tetramer to modify its structure).
Answer:
We made changes especially in section 2 to always use the term heparin-PF4 complexes when describing the binding of heparin-dependent HIT antibodies or HIT-like antibodies to PF4 (in complex with heparin). For the other parts of the text, if not specifically defined as heparin-PF4 complexes we describe reactivity to PF4 alone. A short sentence clarifying this differentiation was also added to the paragraph (page.3 , lines 82-85).
- Page 3, line 94: the differential reactivity of anti-PF4 antibodies developed in VITT, depends on the assay used. It is rather usual with Elisas, and much less frequent in chemiluminescent assays. However, when the right assay conditions are met, (almost) all VITT antibodies react with HPF4, and not only 5%. Interestingly, anti-PF4 antibodies in VITT were identified by using HPF4 Elisas.
Answer:
There is a misunderstanding. The bead-based chemiluminescence assay for HIT recognizes at best 5% of VITT sera. The microtiterplate based ELISAs, however, are highly sensitive (Platton, Sean et al. “Evaluation of laboratory assays for anti-platelet factor 4 antibodies after ChAdOx1 nCOV-19 vaccination.” Journal of thrombosis and haemostasis : JTH vol. 19,8 (2021): 2007-2013. doi:10.1111/jth.15362.) We clarified this in the manuscript (page 3, lines 109-119)
- Page 3, lines 115-119: the presence of antibodies binding to HPF4, and a subgroup binding also binding to PF4 alone, was already reported since 1992 (reference 47), and many times after.
Answer:
Please see answer to question1 in the global comments. The suggestion was followed and proper references to that passage were added (page.3, lines 120-130).
- Page 4, lines 121-123: when the immune stimulus is a complex between PF4 and polyanions (big enough), antibodies developed are like those produced in HIT, even though some are reactive with PF4 alone (epitope spreading).
Answer:
Some further perspectives were added to this paragraph describing the formation of PF4-polyanion complexes that can be recognized by antibodies. A subset of these antibodies may activate platelets after binding to PF4/heparin complexes (causing HIT) or in the absence of heparin possible due to epitope spreading, i.e. in autoimmune-HIT (Nguyen, T-H et al. 2017, doi:10.1038/ncomms14945). The applied changes now may also serve better to bridge to the following paragraph on aHIT. Please see marked canges (page. 4, lines 156-157)
- Page 4, lines 126-130: this feature was also shown in 2000 by extracting different anti HPF4/PF4 antibodies from the same patients with HIT, and separating them in groups of low and high affinity; only those with the highest affinity activated platelets (Amiral J, Pouplard C, Vissac AM, Walenga JM, Jeske W, Gruel Y. Affinity purification of heparin-dependent antibodies to platelet factor 4 developed in heparin-induced thrombocytopenia: biological characteristics and effects on platelet activation. Br J Haematol. 2000 May;109(2):336-41).
Answer:
The reference was added accordingly. We apologize that we overlooked this important previous work.
- Page 6, lines 219-220: this reference (47) from 1992 already reported that some patients with HIT had antibodies binding to both HPF4 and PF4 (figure 1).
Answer:
We applied changes accordingly (page.3, lines 120-130).
- Page 7, lines 272-273: the very different reactivity of HIT and VITT antibodies to HPF4/PF4 excludes the complex formation with a polyanion.
Answer:
Changes were applied accordingly (page.8, lines 305-315).
- Hexon is a candidate for binding PF4 and triggering the immune response, although its binding to PF4 is not that strong; but the immune response first targeted to the adenovirus hexon (or already preexisting from a viral infection) can in very rare cases be extended to PF4 complexed with it (epitope spreading). Lines 319-320 give a possible explanation for that occurrence.
Answer:
The possibility that hexon-PF4 interactions might be triggering an immune response due to neo-epitope formation after previous or pre-existing viral infections was added to the text (page.9, lines 373-374).
- Global presentation and discussion is very interesting and interestingly reported.
Please note that the reference list was updated due to the applied changes.